# GPT4Scene:
# Understand 3D Scenes from Videos with Vision-Language Models

**Zhangyang Qi[1], Zhixiong Zhang[2], Ye Fang[2], Jiaqi Wang[2†], Hengshuang Zhao[1†]**
[1]The University of Hong Kong, [2]Shanghai AI Lab
[†] Corresponding Author
{zyqi,hszhao}@cs.hku.hk, {zhangzhixiong,fangye,wangjiaqi}@pjlab.org.cn

## Abstract

In recent years, 2D Vision-Language Models (VLMs) have made significant strides in image-text understanding tasks. However, their performance in 3D spatial comprehension, which is critical for embodied intelligence, remains limited. Recent advances have leveraged 3D point clouds and multi-view images as inputs, yielding promising results. However, we propose exploring a purely vision-based solution inspired by human perception, which merely relies on visual cues for 3D spatial understanding. This paper empirically investigates the limitations of VLMs in 3D spatial knowledge, revealing that their primary shortcoming lies in the lack of global-local correspondence between the scene and individual frames. To address this, we introduce GPT4Scene, a novel visual prompting paradigm in VLM training and inference that helps build the global-local relationship, significantly improving the 3D spatial understanding of indoor scenes. Specifically, GPT4Scene constructs a Bird's Eye View (BEV) image from the video and marks consistent object IDs across both frames and the BEV image. The model then inputs the concatenated BEV image and video frames with markers. In zero-shot evaluations, GPT4Scene improves performance over closed-source VLMs like GPT-4o. Additionally, we prepare a processed video dataset consisting of 165K text annotation to fine-tune open-source VLMs, achieving state-of-the-art performance on all 3D understanding tasks. Surprisingly, after training with the GPT4Scene paradigm, VLMs consistently improve during inference, even without object marker prompting and BEV image as explicit correspondence. It demonstrates that the proposed paradigm helps VLMs develop an intrinsic ability to understand 3D scenes, which paves the way for a seamless approach to extending VLMs for 3D scene understanding.

## 1 Introduction

3D scene understanding aims to comprehend the overall layout of the surrounding complex environments and the spatial relationships between objects Azuma et al. (2022); Chen et al. (2021; 2020). It plays a crucial role in applications such as embodied intelligence, virtual reality, and smart cities Huang et al. (2024c); Chen et al. (2024b); Chandrasegaran et al. (2024). With the rapid development of Large Language Models Team (2024c); Team & DeepMind (2024); OpenAI (2023); Team (2024d;b), Vision-Language Models have demonstrated impressive performance in image and video understanding Liu et al. (2023; 2024a); Li et al. (2025a); Zhu et al. (2024a); Wang et al. (2025b). Researchers have extended this paradigm to 3D perception by incorporating point clouds, aiming to improve scene understanding Hong et al. (2023); Chen et al. (2024b); Wang et al. (2025c); Fu et al. (2025b); Man et al. (2024); Huang et al. (2024c;a); Kang et al. (2025a).

Recent 3D point LLMs leverage point clouds aligned with LLM features for indoor scene understanding Chen et al. (2024b); Hong et al. (2023). While combining point clouds and images Zhu et al. (2024b); Chen et al. (2024b); Huang et al. (2024a) in Point-Vision-LLM paradigms improves performance through richer visual cues, aligning these modalities with text remains challenging.

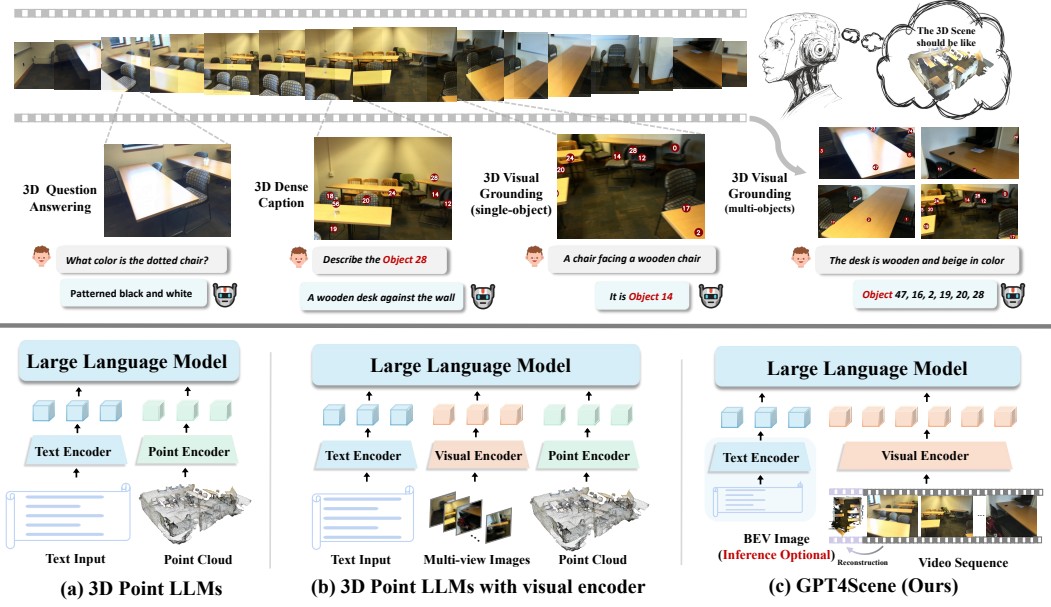

Figure 1: **Overview of GPT4Scene.** GPT4Scene understands 3D scenes and performs tasks like 3D question answering, dense captioning, and visual grounding using only video input. Unlike 3D point LLMs, it relies solely on vision, with global context provided by a BEV image.

This complexity motivates exploration of vision-only solutions, inspired by humans' natural ability to perceive 3D environments without point clouds, offering a direction for efficient scene analysis.

In this work, we aim to leverage pre-trained VLMs without modifying their architecture, maximizing their vision perception capabilities. However, their effectiveness in understanding immersive 3D indoor scenes remains limited. Our analysis shows that directly inputting scene videos into VLMs fails in 3D scene understanding due to two factors: i) the lack of a global scene representation, ii) misalignment between per-frame local observations and their spatial-temporal context.

To address this, we propose GPT4Scene, a framework that enhances VLMs' spatial understanding (see Figure 1). We first perform 3D reconstruction on input videos to generate a Bird's Eye View (BEV) image, offering a comprehensive scene layout. Additionally, we introduce Spatial-Temporal Object markers (STO-markers) in both the BEV image and 2D frames. These markers maintain consistent object IDs across frames (temporal level) and align with the BEV image (spatial level), bridging the global-local relationship. Empirical results show that GPT4Scene remains robust to reconstruction quality and marker accuracy, as it prioritizes learning global-local correspondences over precise geometric reconstructions.

We first explored the effectiveness of GPT4Scene under a training-free approach. Experimental results revealed that it was notably effective for powerful large-scale VLMs such as Qwen2-VL-72B Wang et al. (2025b), as well as closed-source models like GPT-4o OpenAI (2024) and Gemini-1.5-Pro Team (2024a), significantly enhancing their 3D scene understanding capabilities, even reaching levels comparable to previous state-of-the-art point-based methods. However, the improvements were limited on smaller-scale models. For smaller open-source vision-language models (VLMs), we introduce ScanAlign, a multimodal dataset comprising 165K aligned data pairs featuring STO-marker-annotated video frames, BEV images, and textual descriptions. Fine-tuning Qwen2-VL-7B on ScanAlign achieves state-of-the-art performance, showing a substantial relative improvement on 3D question answering (SQA3D) and significantly surpassing the previous SOTA, Chat-scene. The model demonstrates even stronger 3D visual grounding capabilities on Multi3DRef, where it also considerably outperforms Chat-scene. These advancements underscore GPT4Scene's effectiveness in seamlessly enhancing VLMs with strong 3D spatial understanding.

Because of the page limit, the related work is shown in Section F.

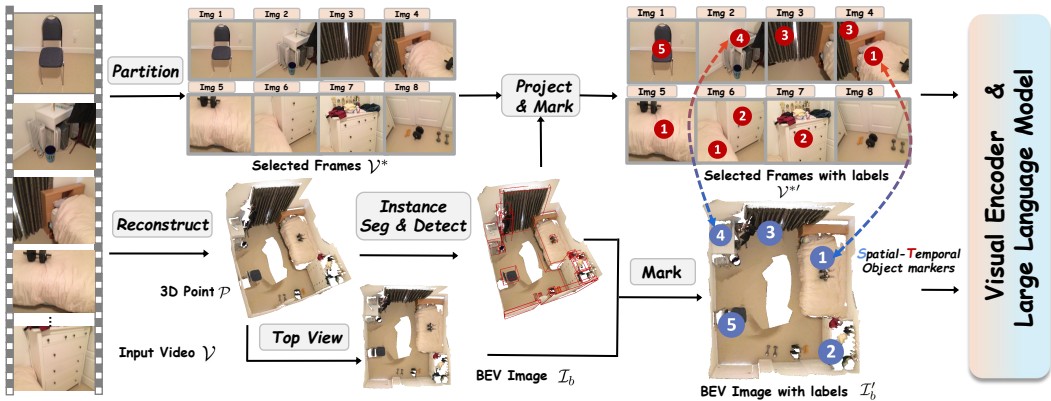

Figure 2: **The framework of GPT4Scene.** A scene video is processed by sampling frames, reconstructing a point cloud, and generating a BEV image. Object locations are detected from the point cloud and projected onto the video frames. The resulting frames and BEV image, enhanced with STO-markers, are inputs for Large Language Model (VLM) training and inference.

Our paper makes these major contributions:

- We introduce GPT4Scene, a framework that enhances Vision-Language Models (VLMs) to comprehend 3D scenes directly from pure vision input.

- We introduce two techniques: i) A 3D BEV image with global context information, and ii) Spatial-Temporal Object markers (STO-markers) for spatial and temporal consistency across BEV image and video frames.

- We introduce ScanAlign, a dataset comprising video frames, BEV images with STO-markers, and text annotations. Fine-tuning VLMs on this dataset significantly improves 3D understanding, even with raw video inputs.

- GPT4Scene demonstrates robust performance in zero-shot and fine-tuning settings, achieving SOTA results across diverse 3D scene understanding tasks.

## 2 METHODOLOGY

We introduce the framework as shown in Subsection 2.1, which improves 3D scene understanding using video inputs. We also explore how zero-shot prompts unlock the potential of large-scale VLMs in Subsection 2.2 and apply fine-tuning for enhanced 3D understanding in Subsection 2.3.

### 2.1 GPT4SCENE FRAMEWORK

**Overview.** Here we introduce GPT4Scene's architecture. As shown in fig. 2, our framework combines global scene layouts with object localization details to enhance VLMs' 3D understanding. Given an input video sequence $\mathcal{V} = \{I_1, \ldots, I_N\}$ captured during indoor scene traversal, we first reconstruct the 3D scene from the complete video sequence, simultaneously generating a Bird's-Eye View (BEV) map and 3D instance segmentation masks. To establish Spatio-Temporal Object markers (STO-markers), we perform uniform frame sampling by selecting $n$ frames at indices: $s_i = \left\lfloor (i-1)\frac{N}{n} \right\rfloor + 1, \quad \forall i \in \{1, \ldots, n\}$ yielding the sampled subset $\mathcal{V}^* = \{I_{s_1}, \ldots, I_{s_n}\}$. The 3D instance segmentation results are then projected onto both the BEV map and these keyframes to enable cross-view object localization.

**Global information: 3D BEV map.** Egocentric videos lack global context, which we address by reconstructing the 3D scene into a point cloud and rendering a bird's-eye view (BEV) image for holistic VLM understanding. Given video $\mathcal{V} = \{I_1, I_2, \ldots, I_N\}$ and camera extrinsics $\mathcal{E} = \{E_1, E_2, \ldots, E_N\}$, we use 3D reconstruction to generate meshes and point clouds:

$$\mathcal{P} = \mathcal{R}\left(\{(I_t, E_t)\}_{t=1}^N\right) \tag{1}$$

$\mathcal{R}(\cdot)$ denotes the reconstruction process, assuming known camera intrinsics. We then generate a BEV image from the global point cloud:

$$\mathcal{I}_b = \mathcal{T}(\mathcal{P}, \ E_{top}) \tag{2}$$

Here, $E_{\text{top}} \in \text{SE}(3)$ represents the camera extrinsic for the top-down view, and $\mathcal{T}(\cdot)$ renders the BEV image. We provide global 3D information to VLMs as top-view images rather than points.

**Local correspondence: STO-markers.** To help VLMs focus on specific objects, we introduce Spatial-Temporal Object markers (STO-markers), ensuring consistency between 2D frames and the 3D BEV image. From the 3D point cloud $\mathcal{P}$ reconstructed from the video $\mathcal{V}$, we apply 3D instance segmentation (e.g., Mask3D) to obtain instance masks $\mathcal{M} = \{M_1, M_2, \ldots M_K\}$, where $K$ is the total number of objects. For the BEV image $\mathcal{I}_b$, we project the 3D segmentation masks $\mathcal{M}$ onto the $xy$ plane and extract the center coordinates of the bounding box, $\boldsymbol{C^{xy}} = \{C_1^{xy}, C_2^{xy}, \ldots, C_K^{xy}\}$. These coordinates are then overlaid onto the BEV image. For selected egocentric video frames $\mathcal{V}^*$, the 3D instance masks $\mathcal{M}$ are projected onto each frame based on its camera pose. For each frame $i$, we extract the centroid of the 2D mask for each object as its 2D marker, $\boldsymbol{C_i^{uv}} = \{C_{i,1}^{uv}, C_{i,2}^{uv}, \ldots, C_{i,K}^{uv}\}$, where $C_{i,k}^{uv}$ is the 2D marker for the $k$-th object in the $i$-th frame. The processed frames and BEV image with markers are defined as follows:

$$\mathcal{V}^{*\prime} = \{\mathcal{F}(I_i, \ \boldsymbol{C_i^{uv}}) \mid i = s_1, s_2, \ldots, s_n\}, \quad \mathcal{I}_b' = \mathcal{F}(\mathcal{I}_b, \ \boldsymbol{C^{xy}}) \tag{3}$$

Here, $\mathcal{F}(\cdot)$ denotes the STO-marker projection operator that jointly augments both the sampled video frames $\mathcal{V}^*$ and BEV image $\mathcal{I}_b$, producing the marked versions $\mathcal{V}^{*\prime}$ and $\mathcal{I}_b'$ respectively. The 2D markers in each frame and their corresponding 3D markers in the BEV image are spatially aligned, ensuring consistency across frames at the object level for temporal coherence.

## 2.2 The Effectiveness of GPT4Scene in a Zero-Shot Setting

Here, 'zero-shot' refers to the process where we take the original video and, through 3D reconstruction and segmentation, generate a marked Bird's-Eye View (BEV) image and project these markers back onto the 2D video frames. **This results in a video with STO-Markers.** Subsequently, the processed video is fed into the VLM to execute the Question Answering task. As shown in Table 1, our approach had limited impact on smaller models but significantly improved performance for large-scale ones (including the Qwen2-VL-72B, GPT-4o, and Gemini-2.5 Pro), matching the Chat-scene Huang et al. (2024a). These findings indicate the limitations of smaller Vision-Language Models (VLMs) in 3D scene understanding with zero-shot prompting alone, which motivated us to propose the ScanAlign dataset and fine-tune open-source models for enhanced performance.

Table 1: **The zero-shot capability of GPT4Scene.** Video + GPT4Scene Inference without Fine-tuning.

| Zero-shot 3D QA | ROUGE@ScanQA | | EM-1@SQA3D | |
|---|---|---|---|---|
| | VID | +4Scene | VID | +4Scene |
| *3D LLM Based Model* | *Pre SOTA* | | *Pre SOTA* | |
| Chat-scene Huang et al. (2024a) | | *41.6* | | *54.6* |
| *Open-sourced VLM Based Model* | | | | |
| Qwen2-VL-2B Wang et al. (2025b) | 29.5 | 30.0 +0.5 | 36.9 | 36.2 -0.7 |
| Qwen2-VL-7B Wang et al. (2025b) | 30.8 | 33.2 +2.4 | 42.1 | 43.1 +1.0 |
| Qwen2-VL-72B Wang et al. (2025b) | 32.1 | 35.1 +3.0 | 41.5 | 44.0 +2.5 |
| *Closed-sourced VLM Based Model* | | | | |
| GPT-4o OpenAI (2024) | 34.2 | 39.3 +5.1 | 42.0 | 44.8 +2.8 |
| Gemini-1.5-Pro Team (2024a) | 35.1 | 39.4 +4.3 | 43.5 | 46.0 +2.5 |

In a zero-shot setting, the model must create a global-local understanding of a 3D scene by fusing local 2D frame features with global BEV (Bird's-Eye View) features. Smaller models, such as Qwen2-VL-2B and Qwen2-VL-7B, are constrained by their parameter scale (typically in the low billions). Their visual encoders and cross-modal fusion capabilities are consequently weaker. They struggle to fully parse the global layout logic from the BEV image (e.g., the relative positions of objects, spatial boundaries) and cannot precisely match the consistency of STO-markers across different frames and the BEV. This leads to a breakdown in the "global-local association." For instance, Results show that the small model Qwen2-VL-2B's zero-shot score actually dropped, a clear sign of its "capability overload. In contrast, large-scale models like Qwen2-VL-72B and GPT-4o possess the architectural complexity to inherently grasp these feature associations, allowing them to form a preliminary 3D cognitive understanding even without any fine-tuning.

## 2.3 Enhancing VLMs with ScanAlign Fine-Tuning

The analysis in the previous section indicates that while zero-shot prompting activates spatial awareness in large-scale models, smaller open-source VLMs require explicit fine-tuning to acquire this capability. To this end, we constructed **ScanAlign**, a large-scale instruction-tuning dataset designed to empower 2D VLMs with 3D understanding.

Distinct from raw data collection, ScanAlign unifies and transforms existing high-quality 3D annotations into a novel visual prompting format. We aggregate 165K text-scene pairs from five mainstream benchmarks based on ScanNet Dai et al. (2017a)—ScanQA Azuma et al. (2022), SQA3D Ma et al. (2023), Scan2Cap Chen et al. (2021), ScanRefer Chen et al. (2020), and Multi3DRef Zhang et al. (2023c). While the text annotations are derived from these prior works, our core contribution lies in **reformatting** these purely geometric data entries (point clouds) into a VLM-compatible format $(\mathcal{V}^{*\prime}, \mathcal{I}_b^\prime, T)$. Specifically, we augment the original labels by generating synchronized egocentric video frames and projecting Spatial-Temporal Object (STO) markers onto both frames and BEV images.

Table 2: **ScanAlign**: Datasets used for training GPT4Scene (Supervised Fine-Tuning),

| Source | Data Type | Task Type | Samples |
|---|---|---|---|
| *Overall Scene-Level* | | | |
| ScanQA | 3D QA | Spatial Relationship
Object Attribute
Existence & Counting | 26K |
| SQA3D | 3D QA | Egocentric Spatial
Navigation
Object Attribute | 26K |
| Total | | | 52K |
| *Object-Level* | | | |
| Scan2cap | 3D Caption | Object Attributes
Spatial Location
State or Affordance | 35K |
| ScanRefer | 3D Grounding | Combined Attributes
Spatial Relations
Unique Attributes | 41K |
| Multi3DRef | 3D Grounding | Group Reference
List-based Refer
Relational Refer | 35K |
| Total | | | 165K |

## 3 Experiments

In this section, we primarily present the experimental results. First, we outline the training details in Subsection 3.1, and subsequently introduce the fine-tuning main results in Subsection 3.2. Finally, Subsection 3.3 details the ablation study, demonstrating the effectiveness of individual components.

### 3.1 Implementation details

Our 3D scene understanding benchmark is based on the ScanNet Dai et al. (2017a) dataset and includes three tasks: 3D question answering (ScanQA Azuma et al. (2022) and SQA3D Ma et al. (2023)), 3D dense captioning (Scan2Cap Chen et al. (2021)), and 3D visual grounding (ScanRefer Chen et al. (2020) and Multi3DRef Zhang et al. (2023c)). Our pipeline begins with 3D scene reconstruction using BundleFusion Dai et al. (2017b) to generate dense point clouds, followed by 3D instance segmentation with Mask3D Schult et al. (2023) to extract object-level masks. The segmented 3D objects are then projected to BEV coordinates, while their corresponding masks are mapped to 2D image frames to establish Spatial-Temporal Object (STO) markers. This comprehensive processing framework enables precise cross-modal alignment essential for both dense captioning and visual grounding tasks. For the experiment, we sample N=32 frames per video (512×490 resolution) for all models. Training is done for one epoch with a base learning rate of 5e-6 and cosine annealing, completing in about 6 hours on 8×A100 GPUs.

### 3.2 Fine-tuning main results

We performed fine-tuning exclusively on ScanAlign: using GPT4Scene as the dataset, we fine-tuned InternVL-3 8B Chen et al. (2024c), Qwen2-VL 7B Wang et al. (2025b), and Qwen2.5-VL 8B Team (2024d), and ultimately achieved state-of-the-art (SOTA) results.

**3D question answering.** The 3D question answering results presented in Table 3 are categorized into three distinct groups: **(1)** conventional task-specific models, **(2)** architectures based on 3D point cloud LLMs, and **(3)** vision-language multimodal LLM systems. Our experimental analysis demonstrates that the baseline Qwen2-VL-7B model without fine-tuning shows constrained capability in 3D QA scenarios. In terms of specific metrics, models fine-tuned using the GPT4Scene framework (based on the ScanAlign dataset) show outstanding performance: Qwen2-VL-7B (GPT4Scene) achieves a BLEU-1 score of 44.4 and a CIDEr score of 96.3 on the ScanQA dataset, along with

Table 3: **Evaluation of 3D question answer on ScanQA Azuma et al. (2022) & SQA3D Ma et al. (2023).**

| 3D Question Answering Methods | Point Encoder | Vision Encoder | ScanQA (val) | | | | | SQA3D (val) | |
|---|---|---|---|---|---|---|---|---|---|
| | | | BLEU-1 | BLEU-4 | METEOR | ROUGE | CIDEr | EM-1 | EM-R1 |
| *Task-Specific Model* | | | | | | | | | |
| ScanQA Azuma et al. (2022) | ✓ | ✗ | 30.2 | 10.1 | 13.1 | 33.3 | 64.9 | - | - |
| SQA3D Ma et al. (2023) | ✓ | ✗ | - | - | - | - | - | 46.6 | - |
| 3D-VLP Jin et al. (2023) | ✓ | ✗ | 30.5 | 11.2 | 13.5 | 34.5 | - | - | - |
| 3D-Vista Zhu et al. (2023) | ✓ | ✗ | - | - | 13.9 | 35.7 | - | 48.5 | - |
| *3D LLM Based Model* | | | | | | | | | |
| Chat-3D Wang et al. (2025c) | ✓ | ✗ | 29.1 | 6.4 | 11.9 | 28.5 | 53.2 | - | - |
| Chat-3D v2 Huang et al. (2024b) | ✓ | ✗ | 38.4 | 7.3 | 16.1 | 40.1 | 77.1 | - | - |
| 3D-LLM Hong et al. (2023) | ✓ | ✓ | 39.3 | 12.0 | 14.5 | 35.7 | 69.4 | - | - |
| LL3DA Chen et al. (2024b) | ✓ | ✗ | - | 13.5 | 15.9 | 37.3 | 76.8 | - | - |
| PQ3D Zhu et al. (2024b) | ✓ | ✓ | - | - | - | - | - | 47.1 | - |
| LEO† Huang et al. (2024c) | ✓ | ✓ | - | 11.5 | 16.2 | 39.3 | 80.0 | 50.0 | 52.4 |
| Grounded-3D-LLM Chen et al. (2025) | ✓ | ✓ | - | 13.4 | - | - | 72.7 | - | - |
| Chat-scene Huang et al. (2024a) | ✓ | ✓ | 43.2 | 14.3 | 18.0 | 41.6 | 87.7 | 54.6 | 57.5 |
| ROSS3D Wang et al. (2025a) | ✗ | ✓ | 49.2 | 17.9 | 20.9 | **50.7** | **107.0** | 63.0 | 65.7 |
| *Vision LLM Based Model* | | | | | | | | | |
| SceneLLM Fu et al. (2025b) | ✗ | ✓ | 43.6 | 12.0 | 16.6 | 40.0 | 80.0 | 54.2 | - |
| LLaVA-3D Zhu et al. (2025) | ✗ | ✓ | - | 14.5 | 20.7 | 50.1 | 91.7 | 55.6 | - |
| Video-3D-LLM Zheng et al. (2025) | ✗ | ✓ | 47.1 | 16.2 | 19.8 | - | 102.1 | 58.6 | - |
| InternVL3-8B (GPT4Scene) | ✗ | ✓ | 45.1 | 16.2 | 19.5 | 47.8 | 96.8 | 61.9 | 64.5 |
| Qwen2-VL-7B (GPT4Scene) | ✗ | ✓ | 44.4 | 15.5 | 18.9 | 46.5 | 96.3 | 60.6 | 63.3 |
| Qwen2.5-VL-7B (GPT4Scene) | ✗ | ✓ | **49.2** | **19.8** | **21.1** | 50.2 | 105.7 | **63.5** | **66.2** |

an EM-1 score of 60.6 and an EM-R1 score of 63.3 on the SQA3D dataset; The more powerful Qwen2.5-VL-7B (GPT4Scene) further improves the BLEU-1 score to 49.2 and the CIDEr score to 105.7 on ScanQA, while reaching an EM-1 score of 63.5 and an EM-R1 score of 66.2 on SQA3D. These models not only significantly outperform the untuned baseline VLMs but also comprehensively outperform the previous SOTA models in the 3D point cloud LLM category (e.g., Chat-scene). This fully demonstrates that GPT4Scene—by providing global context through BEV (Bird's-Eye View) images and establishing spatiotemporal consistency via STO-markers—can effectively enhance the 3D spatial reasoning and QA capabilities of VLMs.

**3D dense caption.** Table 4 presents the primary evaluation for the 3D dense captioning task on the Scan2Cap benchmark. The table contrasts the performance of various models using BLEU-4 and ROUGE scores, evaluated at both a lenient (IoU@0.25) and a strict (IoU@0.5) matching threshold. The key objective is to demonstrate the efficacy of our GPT4Scene framework in enhancing the capability of Vision Language Models (VLMs) for detailed 3D scene description. Our GPT4Scene-integrated Visual LLMs, including Qwen2-VL-7B (GPT4Scene) and Qwen2.5-VL-7B (GPT4Scene), achieve breakthrough performance using only visual inputs (video + BEV images). Specifically, Qwen2.5-VL-7B (GPT4Scene) scores a BLEU-4 of 45.9 and a ROUGE of 67.9 at IoU@0.25, and a BLEU-4 of 44.1 with a ROUGE of 67.1 at IoU@0.5. These results comprehensively outperform the other model classes. This superior performance stems from GPT4Scene's ability to provide global scene context via BEV imagery and maintain spatio-temporal consistency through STO-markers. Consequently, the VLM can precisely ground textual descriptions in 3D scene details without requiring 3D point cloud data, thereby establishing a new state-of-the-art (SOTA) in the field.

**3D visual grounding.** Table 5 presents the critical evaluation for the 3D visual grounding task, which aims to accurately align textual descriptions with their corresponding objects in 3D space. Using the ScanRefer (single/multi-object grounding) and Multi3DRef (combinatorial multi-object grounding) benchmarks, this analysis validates the enhancement of spatial localization capabilities in Vision Language Models (VLMs) by our GPT4Scene framework. Performance is measured by localization accuracy (Acc@0.25 and Acc@0.5) on ScanRefer and the global F1 score (all F1@0.25 and all F1@0.5) on Multi3DRef. Vision LLMs integrated with GPT4Scene, such as InternVL3-

Table 4: **Evaluation of 3D dense caption on Scan2Cap** Chen et al. (2021)**.** Our results outperform those of existing 3D LLM based models.

| 3D Dense Caption | IoU@0.25 | | IoU@0.5 | |
|---|---|---|---|---|
| Methods | BLEU-4 | ROUGE | BLEU-4 | ROUGE |
| *Task-Specific Model* | | | | |
| Scan2Cap Chen et al. (2021) | 34.2 | 55.3 | 23.3 | 44.5 |
| 3DJCG Cai et al. (2022) | 40.2 | 59.2 | 31.0 | 50.8 |
| X-Trans2Cap Yuan et al. (2022) | 35.7 | 54.7 | 25.1 | 45.3 |
| 3D-VisTA Zhu et al. (2023) | 36.5 | 57.6 | 34.0 | 54.3 |
| Vote2Cap-DETR Chen et al. (2023b) | 39.3 | 59.3 | 34.5 | 54.4 |
| *3D LLM Based Model* | | | | |
| LL3DA Chen et al. (2024b) | 41.4 | 59.5 | 36.8 | 55.1 |
| PQ3D Zhu et al. (2024b) | – | – | 36.0 | - |
| LEO Huang et al. (2024c) | – | – | 36.9 | 57.8 |
| Chat-scene Huang et al. (2024a) | 38.2 | 60.6 | 36.3 | 58.1 |
| Grounded 3D-LLM Chen et al. (2025) | – | – | 35.5 | - |
| Robin3D Kang et al. (2025a) | – | – | 38.4 | – |
| Ross3D Wang et al. (2025a) | – | – | 43.4 | 66.9 |
| *Vision LLM Based Model* | | | | |
| LLaVA-3D Zhu et al. (2025) | – | – | 41.1 | 63.4 |
| Video-3D-LLM Zheng et al. (2025) | – | – | 41.3 | - |
| InternVL3-8B (GPT4Scene) | 44.1 | 63.1 | 41.4 | 60.3 |
| Qwen2-VL-7B (GPT4Scene) | 43.1 | 61.9 | 40.6 | 59.3 |
| Qwen2.5-VL-7B (GPT4Scene) | **45.9** | **67.9** | **44.1** | **67.1** |

Table 5: **Evaluation of 3D visual grounding on ScanRefer** Chen et al. (2020) **and Multi3DRef** Zhang et al. (2023c)**.** Our method reaches state-of-the-art performance over all methods for the 3D visual grounding task.

| 3D Visual Grounding | ScanRefer | | Multi3DRef | |
|---|---|---|---|---|
| Methods | Acc@0.25 | Acc@0.50 | all F1@0.25 | all F1@0.50 |
| *Task-Specific Model* | | | | |
| 3DVG-Transformer Zhao et al. (2021) | 47.6 | 34.7 | – | 25.5 |
| 3DJCG Cai et al. (2022) | 49.6 | 37.3 | – | 26.6 |
| D3Net Chen et al. (2022a) | – | 37.9 | – | 32.2 |
| M3DRef-CLIP Zhang et al. (2023c) | 51.9 | 44.7 | 42.8 | 38.4 |
| *3D LLM Based Model* | | | | |
| 3D-LLM Hong et al. (2023) | 30.3 | - | - | - |
| Grounded 3D-LLM Chen et al. (2025) | 47.9 | 44.1 | 45.2 | 40.6 |
| Chat-scene Huang et al. (2024a) | 55.5 | 50.2 | 57.1 | 52.4 |
| Ross3D Wang et al. (2025a) | 61.1 | 54.4 | 59.6 | 54.3 |
| *Vision LLM Based Model* | | | | |
| LLaVA-3D Zhu et al. (2025) | 54.1 | 42.4 | - | - |
| Video-3D-LLM Zheng et al. (2025) | 58.1 | 51.7 | 58.0 | 52.7 |
| InternVL3-8B (GPT4Scene) | 63.4 | 57.7 | 65.5 | 60.7 |
| Qwen2-VL-7B (GPT4Scene) | 62.6 | 57.0 | 64.5 | 59.8 |
| Qwen2.5-VL-7B (GPT4Scene) | 65.6 | 59.5 | 67.3 | 62.8 |

8B (GPT4Scene) and Qwen2-VL-7B (GPT4Scene), demonstrate breakthrough performance using only visual inputs (video + BEV imagery). For instance, Qwen2-VL-7B (GPT4Scene) achieves an Acc@0.25 of 62.6 on ScanRefer and an all F1@0.25 of 64.5 on Multi3DRef, surpassing the previous SOTA, Chat-scene, by 7.1 and 7.4 points, respectively. Furthermore, our improved Qwen2.5-VL-7B (GPT4Scene) model sets a new state-of-the-art, elevating these scores to 65.6 and 67.3.

**GPT score: new evaluation way.** We propose a novel GPT Score for 3D QA assessment, using the state-of-the-art 3D LLM Chat-scene Huang et al. (2024a) as the baseline. For our comparison, we selected a total of 1,000 questions from the ScanQA dataset, distributed across six distinct categories: Spatial Relationship, Object Attribute, Existence & Counting, State or Affordance, Group Reference, and Relational Refer. We then prompted GPT-4o OpenAI (2024) to compare the responses from our GPT4Scene-finetuned model against those from Chat-scene. The evaluation revealed a clear pattern: GPT4Scene demonstrates a significant performance advantage in object-centric categories, particularly excelling in tasks related to Relational Refer and Existence & Counting. Conversely, the performance gap in the Spatial Relationship category was not significant, indicating comparable capabilities in that area. These findings strongly suggest that the core strengths of GPT4Scene lie in its enhanced understanding and processing at the object level.

## 3.3 ABLATION STUDY

In this section, we conduct ablation studies to validate the effectiveness of GPT4Scene. First, we evaluate its robustness, including performance on small objects, followed by analyzing the robustness of STO-markers and reconstruction quality. Next, we perform module-wise ablation to assess individual components. Finally, we investigate the impact of frame intervals and resolution settings.

**The effectiveness during training and inference.** Table 6 presents an ablation study on the GPT4Scene framework, analyzing the contribution of its components by selectively enabling them during training and inference. The study's key finding is that the fine-tuning process with GPT4Scene is the most critical driver of performance. A model trained this way maintains strong results even without the special inference components (BEV images and STO-markers), demonstrating that it acquires an intrinsic 3D understanding. While training is fundamental, the inference components provide a further enhancement, pushing the model to its peak performance when combined. This study clearly validates the dual effectiveness of the GPT4Scene framework in both empowering the model through training and enhancing its capabilities at inference.

Table 6: **Ablation study on the Efficacy of GPT4Scene.** (1) on fully fine-tuned models with GPT4Scene; (2) on pure-video fine-tuned models; (3) in a zero-shot setting without training. Exp conducted on Qwen2-VL-7B.

| GPT4Scene Train | GPT4Scene Eval | METEOR | ROUGE | CIDEr |
|---|---|---|---|---|
| ✓ | ✓ | **18.9** | **46.5** | **96.3** |
| ✓ | ✗ | 18.6 | 45.9 | 95.4 |
| (Pure Video SFT) ✗ | ✓ | 17.3 | 43.5 | 88.2 |
| (Pure Video SFT) ✗ | ✗ | 16.7 | 42.1 | 85.9 |
| (No training) ✗ | ✓ | 14.1 | 33.2 | 68.7 |
| (No training) ✗ | ✗ | 12.4 | 30.8 | 64.7 |

Table 7: **Ablation Study on BEV Reconstruction Quality.** The quality of BEV reconstruction has a negligible impact on QA performance, since the BEV mainly offers a global overview of the scene.

| BEV Reconstruction Quality | ScanQA ROUGE | SQA3D EM-1 |
|---|---|---|
| BundleFusion (Baseline) | 46.5 | 60.6 |
| SLAM3R, 50-frame intervals | 46.1 | 58.8 |
| 100-frame intervals | 46.7 | **61.3** |
| 200-frame intervals | 45.8 | 59.1 |
| GS-SLAM | 46.3 | 60.7 |
| MAST3R-SLAM | **47.0** | 59.5 |

Table 8: **Ablation on the size of objects.** Comprehension is most effective for large objects, and diminishes for smaller ones.

| Connector | METEOR | ROUGE | CIDEr |
|---|---|---|---|
| All objects | 18.9 | 46.5 | 96.3 |
| Small (e.g., alarm clock) | 17.2 | 42.8 | 91.5 |
| Middle (e.g., chair) | 18.7 | 46.6 | 95.8 |
| Large (e.g., cabinet) | **19.3** | **47.3** | **97.5** |

Table 9: **Ablation study on the BEV and STO-Markers.** Both the BEV Image and STO-markers improve understanding. Performance peaks at an optimal STO-marker size before degrading.

| ScanQA | w/i BEV Image | | | w/o BEV Image | | |
|---|---|---|---|---|---|---|
| | METEOR | ROUGE | CIDEr | METEOR | ROUGE | CIDEr |
| w/o STO-markers | $17.5_{\downarrow1.4}$ | $43.1_{\downarrow3.4}$ | $90.4_{\downarrow5.9}$ | $16.0_{\downarrow2.9}$ | $40.0_{\downarrow6.5}$ | $84.1_{\downarrow12.2}$ |
| - size: 30, STO | $18.2_{\downarrow0.7}$ | $45.3_{\downarrow1.2}$ | $93.8_{\downarrow2.5}$ | $16.7_{\downarrow2.2}$ | $42.2_{\downarrow4.3}$ | $87.5_{\downarrow8.8}$ |
| - size: 40, STO | $\mathbf{18.9}_{\downarrow0.0}$ | $\mathbf{46.5}_{\downarrow0.0}$ | $\mathbf{96.3}_{\downarrow0.0}$ | $17.4_{\downarrow1.5}$ | $43.3_{\downarrow3.2}$ | $90.1_{\downarrow6.2}$ |
| - size: 50, STO | $18.0_{\downarrow0.9}$ | $44.9_{\downarrow1.6}$ | $93.1_{\downarrow3.2}$ | $16.5_{\downarrow2.4}$ | $41.8_{\downarrow4.7}$ | $86.9_{\downarrow9.4}$ |

**Ablation Study on BEV Reconstruction Quality.** This ablation study in Table 7 investigates the impact of BEV (Bird's-Eye View) reconstruction quality on model performance. The central hypothesis is that the BEV's primary role is to provide global scene context, not precise geometric detail. The experiment compared various reconstruction methods (e.g., the baseline BundleFusion, SLAM3R at different frame rates) on the ScanQA and SQA3D benchmarks. The results show that despite variations in reconstruction quality, key performance metrics like ROUGE and EM-1 remained stable with only negligible fluctuations. This strongly confirms that the GPT4Scene framework is robust to the geometric precision of the BEV map, depending on it for overall layout rather than fine-grained accuracy.

**Ablation on the size of objects.** This ablation study in Table 8 analyzes how object size impacts the model's 3D understanding capabilities by categorizing objects into small, medium, and large. The results reveal a clear positive correlation between object size and performance: the model understands large objects best, performs near the average for medium objects, and is least effective with small objects. This is because larger objects provide more prominent visual features and clearer spatial footprints in video frames and BEV maps, making them easier to ground. Nevertheless, the model maintains a strong baseline performance even on small objects, confirming the overall effectiveness and robustness of the GPT4Scene framework across various object scales.

**Ablation study on the BEV and STO-Markers.** This ablation study in Table 9 provides a detailed analysis of GPT4Scene's two core components: the BEV image and STO-Markers. The results yield two critical conclusions. First, both the BEV image and STO-Markers are essential for performance and demonstrate a clear synergistic effect. The data shows that removing either component—for instance, removing STO-Markers while keeping the BEV image, or vice versa—causes a significant decline across all key metrics (METEOR, ROUGE, and CIDEr). This confirms that the global scene context provided by the BEV and the spatio-temporal consistency established by STO-Markers are both indispensable for high-fidelity 3D comprehension. Second, the study reveals that there is an optimal size for the STO-Markers. Performance peaked when the marker size was set to 40, while smaller (30) or larger (50) sizes led to a drop in performance. This suggests that the marker scale must be properly adapted to the visual features of objects within the scene to maximize its effectiveness in linking global and local information.

**Ablation on the number of input frames and image resolution.** Figure 4 visualizes an ablation study that assesses the impact of two key parameters—the number of input frames and image resolution—on model performance across different 3D understanding tasks. The experiment compares two core tasks: 3D Question Answering (QA) and 3D Visual Grounding. The central finding is that these parameters affect the two tasks very differently. Both higher image resolution and an increased number of input frames provide a substantial performance boost for the 3D Visual Grounding task, which relies on precise object localization. In contrast, the same parameter upgrades yield only

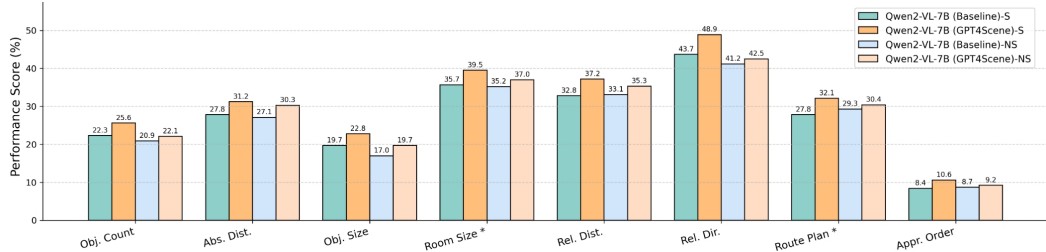

Figure 3: **Evaluation of GPT4Scene on the VSIBench.** 'S' denotes performance on ScanNet, while 'NS' refers to the ARKitScenes dataset. The results indicate that GPT4Scene enhances spatial intelligence.

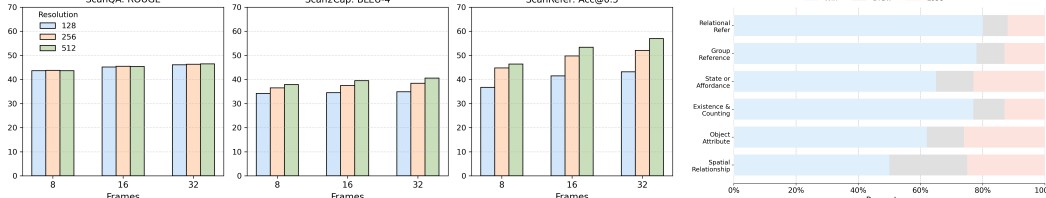

Figure 4: **Ablation on the number of input frames and image resolution.** We find that these parameters have the most significant impact on the Grounding task and the least on question answering.

Figure 5: **GPT-Score evaluation.** GPT4Scene holds an advantage on object-level tasks than Chat-scene.

minimal improvements for the 3D QA task, which involves more general scene comprehension. This result leads to a clear conclusion: for tasks like Visual Grounding, prioritizing high-resolution, multi-frame input is crucial for achieving the best results. For tasks like 3D QA, a more balanced configuration can be used to optimize for efficiency without a significant loss in performance.

### 3.4 EALUATION ON SPATIAL INTELLIGENCE: VSIBENCH

We have completed scene understanding and conducted tests on spatial reasoning. Figure 3 presents a visual analysis of the GPT4Scene framework's performance on the VSIBench, a benchmark designed specifically for evaluating spatial intelligence. The primary goal of this evaluation is to validate how effectively GPT4Scene enhances the spatial reasoning capabilities of Vision Language Models (VLMs). The experiments are conducted across two different datasets, ScanNet ("S") and ARKitScenes ("NS"), to test the framework's robustness in various types of 3D environments. The key results visualized in the chart show that models integrated with GPT4Scene consistently outperform their baseline counterparts across both datasets. This superiority is particularly evident in tasks that demand a high degree of spatial awareness, such as judging spatial relationships between objects and understanding the overall layout of a scene. This outcome provides strong evidence for the effectiveness of GPT4Scene's core design principles. By providing global scene context through BEV images and establishing spatio-temporal consistency with STO-markers, the framework successfully empowers VLMs to overcome their previous limitations, thereby significantly strengthening their cognitive and reasoning abilities within complex 3D spaces.

### 4 CONCLUSION

We propose GPT4Scene, a framework enhancing VLMs to understand 3D scenes directly from visual inputs. By reconstructing 3D point clouds for Bird's Eye View (BEV) images and aligning video frames with spatial-temporal object (STO) markers, we enable global-local scene comprehension. GPT4Scene achieves state-of-the-art 3D QA performance with zero-shot GPT-4o and by fine-tuning smaller VLMs using our ScanAlign dataset. These fine-tuned models even excel with raw video inputs, proving effective 3D understanding. Despite relying on point cloud annotations for marker generation due to benchmark constraints, we aim to address this by generating STO-markers from video segmentation in future work.

**Acknowledgement.** This work is supported by the National Natural Science Foundation of China (No. 62422606, 62441615) and Hong Kong Research Grant Council General Research Fund (No. 17213925).

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

## A   LLM USAGE STATEMENT

During the preparation of this manuscript, we utilized a Large Language Model (LLM) as an assistive tool. The primary role of the LLM was for language polishing, which included improving grammar, enhancing the clarity and fluency of the text, and ensuring consistent use of technical terminology. All core research ideas, methodologies, experimental designs, results, and conclusions were conceived and formulated exclusively by the human authors. The LLM did not contribute to any of the substantive scientific aspects of this work. We have carefully reviewed and edited all text modified with the assistance of the LLM and take full responsibility for the final content of.

## B   PROMPTS OF CLOSED-SOURCE VLMS

Here, we present the prompts used by GPT4Scene in the closed-source VLMs (GPT-4o), as illustrated in Figure 6. The process begins with a general **system prompt**, which outlines the overview of two images provided as input. The first image is a stitched 2D view captured from a video, with dimensions of $2 \times 4$. The second image represents a **BEV** (Bird's Eye View). Subsequently, we perform evaluations across various tasks and benchmarks, with each benchmark associated with a specific prompt. We take the 3D question-answering task on **ScanQA** as an example. The benchmark prompt consists of three parts:

1. **Important Guidelines:** It clarifies that while we provide object IDs for reference, they cannot be directly used when answering questions. Additionally, it specifies adapting the response style to match that of ScanQA. Since ScanQA's responses are typically short single words, we aim to keep the answers concise within the benchmark prompt, targeting 1-5 words.

2. **Answer Format:** In this part, we use a standardized regularized format to structure the answers, which helps improve accuracy when addressing questions.

3. **Examples:** we include two example cases.

Our **zero-shot prompting process** is illustrated in the bottom-left corner Figure 6. The system prompt and ScanQA prompt are used as the system message. In the user message, we input the stitched image and BEV image. Finally, the query message includes the question. The responses generated through this process require further refinement, as depicted in the bottom-right corner of Figure 6. First, we remove the regularized formatting from the answers. Next, we clean the answers by addressing singular/plural forms and case sensitivity. This final step ensures that we obtain the refined answers.

## C   2D MULTI-MODAL BENCHMARK

We tested the fine-tuned *Qwen2-VL-7B (GPT4Scene)* model on 2D image and video multimodal large models. Table 10 shows the results of MVBench. As we can see, our model shows improvement for the object and action metrics, indicating that the model fine-tuned with ScanAlign is better at handling spatial variations and the information of objects in the scene. Table 11 presents the results on other benchmarks, where we can observe that after our fine-tuning, the model's ability to understand images and videos did not decline significantly. It demonstrates the effectiveness of ScanAlign.

## D   QUALITATIVE RESULTS

Figures 7 to 10 present our qualitative results obtained from Qwen2-VL after fine-tuning on ScanAlign. Figures 7 and 8 demonstrate 3D question answering performance using only unannotated video inputs as context. Our observations indicate the model's capability to effectively address queries from both ScanQA Azuma et al. (2022) and SQA3D Ma et al. (2023) benchmarks. Meanwhile, Figures 9 and 10 exhibit the model's competence in 3D dense captioning and 3D visual grounding tasks requiring annotated inputs. These outcomes substantiate that our framework can generate precise responses through visual information processing alone, without dependency

| Task type | Model | |
|---|---|---|
| | Qwen2-VL | Ours |
| Action Sequence | 85.5 | 82.0 |
| Action Prediction | 69.5 | **70.5** |
| Action Antonym | 83.0 | **86.0** |
| Fine-grained Action | 51.5 | 51.5 |
| Unexpected Action | 82.0 | 78.0 |
| Object Existence | 87.5 | **88.5** |
| Object Interaction | 82.0 | 81.5 |
| Object Shuffle | 41.0 | **45.0** |
| Moving Direction | 42.0 | 40.0 |
| Action Localization | 65.0 | **66.5** |
| Scene Transition | 93.5 | **94.0** |
| Action Count | 47.5 | 43.5 |
| Moving Count | 69.5 | 71.5 |
| Moving Attribute | 90.0 | 88.5 |
| State Change | 48.0 | **49.0** |
| Fine-grained Pose | 63.0 | 63.5 |
| Character Order | 74.5 | 71.0 |
| Egocentric Navigation | 39.5 | **41.5** |
| Episodic Reasoning | 47.0 | 47.0 |
| Counterfactual Inference | 62.5 | **65.5** |
| **Avg** | **66.2** | **66.225** |

Table 10: **The result of MVBench Li et al. (2024a).** After fine-tuning with ScanAlign in GPT4Scene, our model shows improved 2D understanding, particularly in object and action metrics.

| Benchmark | Model | |
|---|---|---|
| | Qwen2-VL | Ours |
| MMBench-EN$_{val}$ Liu et al. (2024c) | 82.4 | 81.2 |
| MMBench-CN$_{val}$ Liu et al. (2024c) | 81.7 | 79.9 |
| MMStar Chen et al. (2024a) | 60.7 | 57.6 |
| RealWorldQA xAI (2024) | 70.1 | 68.5 |
| Video-MME Fu et al. (2025a) | 59.8 | 58.4 |

Table 11: **The result of 2D Multi-modal Benchmark.** After fine-tuning with ScanAlign in GPT4Scene, our model's 2D understanding capabilities did not decline.

on 3D point cloud inputs. Here we emphasize that post-training with the GPT4Scene framework, the model achieves accurate QA performance using only pure video input, thereby demonstrating GPT4Scene's effectiveness in enhancing visual comprehension capabilities.

# E FULL QUANTITIVE RESULTS

Here, we present the complete metrics for all five benchmarks. Table 12 and Table 13 show results for ScanQA Azuma et al. (2022) and SQA3D Ma et al. (2023). Table 14 provides the full results for

| Methods | EM-1 | BLEU-n Metrics | | | | Language Generation Metrics | | |
|---|---|---|---|---|---|---|---|---|
| | | BLEU-1 | BLEU-2 | BLEU-3 | BLEU-4 | ROUGE | METEOR | CIDEr |
| *Task-Specific Model* | | | | | | | | |
| ScanQA Azuma et al. (2022) | 21.1 | 30.2 | 20.4 | 15.1 | 10.1 | 33.3 | 13.1 | 64.9 |
| 3D-VLP Jin et al. (2023) | 21.7 | 30.5 | 21.3 | 16.7 | 11.2 | 34.5 | 13.5 | 67.0 |
| 3D-Vista Zhu et al. (2023) | – | – | – | – | 13.9 | 35.7 | – | – |
| *3D LLM Based Model* | | | | | | | | |
| 3D-LLM Hong et al. (2023) | 20.5 | 39.3 | 25.2 | 18.4 | 12.0 | 35.7 | 14.5 | 69.4 |
| LL3DA Chen et al. (2024b) | – | – | – | – | 13.5 | 37.3 | 15.9 | 76.8 |
| LEO Huang et al. (2024c) | 24.5 | – | – | – | 11.5 | 39.3 | 16.2 | 80.0 |
| Scene-LLM Fu et al. (2025b) | 27.2 | 43.6 | 26.8 | 19.1 | 12.0 | 40.0 | 16.6 | 80.0 |
| Chat-scene Huang et al. (2024a) | 21.6 | 43.2 | 29.1 | 20.6 | 14.3 | 41.6 | 18.0 | 87.7 |
| *Vision LLM Based Model* | | | | | | | | |
| InternVL3-8B (GPT4Scene) | 29.5 | 45.1 | 30.7 | 22.7 | 16.2 | 47.8 | 19.5 | 96.8 |
| Qwen2-VL-7B (GPT4Scene) | 28.2 | 44.4 | 30.3 | 22.3 | 15.5 | 46.5 | 18.9 | 96.3 |
| Qwen2.5-VL-7B (GPT4Scene) | 31.2 | 49.2 | 38.2 | 28.3 | 19.8 | 50.2 | 21.1 | 105.7 |

Table 12: **Full Evaluation of 3D Question Answering on ScanQA Azuma et al. (2022).**

| Methods | Test Set | | | | | | Avg.(EM-1) | EM-R1 |
|---|---|---|---|---|---|---|---|---|
| | What | Is | How | Can | Which | Others | | |
| *Task-Specific Model* | | | | | | | | |
| ClipBERT Ma et al. (2023) | 30.2 | 60.1 | 38.7 | 63.3 | 42.5 | 42.7 | 43.3 | – |
| SQA3D Ma et al. (2023) | 31.6 | 63.8 | 46.0 | 69.5 | 43.9 | 45.3 | 46.6 | – |
| 3D-VisTA Zhu et al. (2023) | 34.8 | 63.3 | 45.4 | 69.8 | 47.2 | 48.1 | 48.5 | – |
| *3D LLM Based Model* | | | | | | | | |
| PQ3D Zhu et al. (2024b) | 37.1 | 61.3 | 44.5 | 60.9 | 47.0 | 45.1 | 47.1 | – |
| LEO Huang et al. (2024c) | – | – | – | – | – | – | 50.0 | 52.4 |
| Scene-LLM Fu et al. (2025b) | 40.9 | 69.1 | 45.0 | **70.8** | 47.2 | 52.3 | 54.2 | – |
| Chat-scene Huang et al. (2024a) | 45.4 | 67.0 | 52.0 | 69.5 | 49.9 | 55.0 | 54.6 | 57.5 |
| *Vision LLM Based Model* | | | | | | | | |
| InternVL3-8B (GPT4Scene) | 58.0 | 71.9 | 53.0 | 70.8 | 55.3 | 62.4 | 61.9 | 64.5 |
| Qwen2-VL-7B (GPT4Scene) | 55.9 | 69.9 | 50.8 | 68.7 | 53.3 | 60.4 | 59.4 | 62.4 |
| Qwen2.5-VL-7B (GPT4Scene) | 60.1 | 72.5 | 55.5 | 71.8 | 56.3 | 64.0 | 63.5 | 66.2 |

Table 13: **Full Evaluation of 3D Question Answering on SQA3D Ma et al. (2023).**

| Methods | IoU@0.25 | | | | IoU@0.5 | | | |
|---|---|---|---|---|---|---|---|---|
| | CIDEr | BLEU-4 | METEOR | ROUGE | CIDEr | BLEU-4 | METEOR | ROUGE |
| *Task-Specific Model* | | | | | | | | |
| Scan2Cap Chen et al. (2021) | 56.8 | 34.2 | 26.3 | 55.3 | 39.1 | 23.3 | 22.0 | 44.5 |
| 3DJCG Cai et al. (2022) | 64.7 | 40.2 | 27.7 | 59.2 | 49.5 | 31.0 | 24.2 | 50.8 |
| X-Trans2Cap Yuan et al. (2022) | 61.8 | 35.7 | 26.6 | 54.7 | 43.9 | 25.1 | 22.5 | 45.3 |
| D3Net Chen et al. (2022a) | – | – | – | – | 62.6 | 35.7 | 25.7 | 53.9 |
| 3D-VLP Jin et al. (2023) | 70.7 | 41.0 | 28.1 | 59.7 | 54.9 | 32.3 | 24.8 | 51.5 |
| Vote2Cap-DETR Chen et al. (2023b) | 71.5 | 39.3 | 28.3 | 59.3 | 62.6 | 35.7 | 25.7 | 53.9 |
| 3D-VisTA Zhu et al. (2023) | 71.0 | 36.5 | 28.4 | 57.6 | 66.9 | 34.0 | 27.1 | 54.3 |
| *3D LLM Based Model* | | | | | | | | |
| LL3DA Chen et al. (2024b) | 74.2 | 41.4 | 27.8 | 59.5 | 65.2 | 36.8 | 26.0 | 55.1 |
| LEO Huang et al. (2024c) | – | – | – | – | 68.4 | 36.9 | 27.7 | 57.8 |
| Chat-scene Huang et al. (2024a) | 81.9 | 38.2 | 29.0 | 60.6 | 77.2 | 36.3 | 28.0 | 58.1 |
| Robin3D Kang et al. (2025a) | – | – | – | – | **87.2** | 38.4 | – | – |
| *Vision LLM Based Model* | | | | | | | | |
| InternVL3-8B (GPT4Scene) | 93.5 | 44.1 | 29.8 | 63.1 | 88.1 | 41.4 | 28.7 | 60.3 |
| Qwen2-VL-7B (GPT4Scene) | 91.7 | 43.1 | 29.3 | 61.9 | 86.3 | 40.6 | 28.2 | 59.3 |
| Qwen2.5-VL-7B (GPT4Scene) | 98.2 | 45.9 | 31.5 | 67.9 | 92.5 | 44.1 | 30.1 | 67.1 |

Table 14: **Full Evaluation of 3D Dense Caption on Scan2Cap Chen et al. (2021).**

Scan2Cap Chen et al. (2021), while Table 15 and Table 16 present the results for ScanRefer Chen et al. (2020) and Multi3DRef Zhang et al. (2023c). Our model significantly improves across all benchmarks, highlighting that only pure vision language models can understand 3D scenes effectively.

| Methods | Unique | | Multiple | | Overall | |
|---|---|---|---|---|---|---|
| | Acc@0.25 | Acc@0.5 | Acc@0.25 | Acc@0.5 | Acc@0.25 | Acc@0.5 |
| *Task-Specific Model* | | | | | | |
| ScanRefer Chen et al. (2020) | 76.3 | 53.5 | 32.7 | 21.1 | 41.2 | 27.4 |
| TGNN Huang et al. (2021) | 68.6 | 56.8 | 29.8 | 23.2 | 37.4 | 29.7 |
| SAT Yuan et al. (2022) | 73.2 | 50.8 | 37.6 | 25.2 | 44.5 | 30.1 |
| InstanceRefer Yuan et al. (2021) | 75.7 | 64.7 | 29.4 | 23.0 | 38.4 | 31.1 |
| 3DVG-Transformer Zhao et al. (2021) | 81.9 | 60.6 | 39.3 | 28.4 | 47.6 | 34.7 |
| MVT Huang et al. (2022a) | 77.7 | 66.4 | 31.9 | 25.3 | 40.8 | 33.3 |
| 3D-SPS Luo et al. (2022) | 84.1 | 66.7 | 40.3 | 29.8 | 48.8 | 37.0 |
| ViL3DRel Chen et al. (2022d) | 81.6 | 68.6 | 40.3 | 30.7 | 47.9 | 37.7 |
| 3DJCG Cai et al. (2022) | 83.5 | 64.3 | 41.4 | 30.8 | 49.6 | 37.3 |
| D3Net Chen et al. (2022a) | – | 72.0 | – | 30.1 | – | 37.9 |
| BUTD-DETR Jain et al. (2022) | 84.2 | 66.3 | 46.6 | 35.1 | 52.2 | 39.8 |
| HAM Chen et al. (2022b) | 79.2 | 67.9 | 41.5 | 34.0 | 48.8 | 40.6 |
| 3DRP-Net Wang et al. (2023) | 83.1 | 67.7 | 42.1 | 32.0 | 50.1 | 38.9 |
| 3D-VLP Jin et al. (2023) | 84.2 | 64.6 | 43.5 | 33.4 | 51.4 | 39.5 |
| EDA Wu et al. (2023) | 85.8 | 68.6 | **49.1** | 37.6 | 54.6 | 42.3 |
| M3DRef-CLIP Zhang et al. (2023c) | 85.3 | 77.2 | 43.8 | 36.8 | 51.9 | 44.7 |
| 3D-VisTA Zhu et al. (2023) | 81.6 | 75.1 | 43.7 | 39.1 | 50.6 | 45.8 |
| ConcreteNet Unal et al. (2024) | 86.4 | 82.1 | 42.4 | 38.4 | 50.6 | 46.5 |
| DOrA Wu et al. (2025) | – | – | – | – | 52.8 | 44.8 |
| *3D LLM Based Model* | | | | | | |
| Chat-scene Huang et al. (2024a) | 89.6 | 82.5 | 47.8 | 42.9 | 55.5 | 50.2 |
| Robin3D Kang et al. (2025a) | – | – | – | – | 60.8 | 55.1 |
| *Vision LLM Based Model* | | | | | | |
| InternVL3-8B (GPT4Scene) | 90.8 | 84.5 | 57.2 | 51.8 | 63.4 | 57.7 |
| Qwen2-VL-7B (GPT4Scene) | 90.3 | 83.7 | 56.4 | 50.9 | 62.6 | 57.0 |
| Qwen2.5-VL-7B (GPT4Scene) | 91.5 | 85.8 | 59.2 | 53.7 | 65.6 | 59.5 |

Table 15: **Full Evaluation of 3D Visual Grounding on ScanRefer Chen et al. (2020).**

| Methods | ZT w/o D | ZT w/ D | ST w/o D | | ST w/ D | | MT | | ALL | |
|---|---|---|---|---|---|---|---|---|---|---|
| | F1 | F1 | F1@0.25 | F1@0.5 | F1@0.25 | F1@0.5 | F1@0.25 | F1@0.5 | F1@0.25 | F1@0.5 |
| *Task-Specific Model* | | | | | | | | | | |
| 3DVG-Trans+ Zhao et al. (2021) | 87.1 | 45.8 | – | 27.5 | – | 16.7 | – | 26.5 | – | 25.5 |
| D3Net (Grounding) Chen et al. (2022a) | 81.6 | 32.5 | – | 38.6 | – | 23.3 | – | 35.0 | – | 32.2 |
| 3DJCG (Grounding) Cai et al. (2022) | 94.1 | 66.9 | – | 26.0 | – | 16.7 | – | 26.2 | – | 26.6 |
| M3DRef-CLIP Zhang et al. (2023c) | 81.8 | 39.4 | 53.5 | 47.8 | 34.6 | 30.6 | 43.6 | 37.9 | 42.8 | 38.4 |
| *3D LLM Based Model* | | | | | | | | | | |
| Chat-scene Huang et al. (2024a) | 90.3 | 62.6 | 82.9 | 75.9 | 49.1 | 44.5 | 45.7 | 41.1 | 57.1 | 52.4 |
| *Vision LLM Based Model* | | | | | | | | | | |
| InternVL3-8B (GPT4Scene) | 97.6 | 85.1 | 85.5 | 78.3 | 60.8 | 55.9 | 49.2 | 45.3 | 65.5 | 60.7 |
| Qwen2-VL-7B (GPT4Scene) | 97.4 | 84.4 | 85.0 | 77.7 | 59.9 | 55.1 | 48.6 | 44.6 | 64.5 | 59.8 |
| Qwen2.5-VL-7B (GPT4Scene) | 98.0 | 86.5 | 86.8 | 79.9 | 62.5 | 57.8 | 51.0 | 47.1 | 67.3 | 62.8 |

Table 16: **Full Evaluation of 3D Visual Grounding on Multi3DRef Zhang et al. (2023c).**

---

**System Prompt**

You are a 3D indoor scene assistant. We provide **a labeled 2D image** and **a labeled Bird's Eye View (BEV) image** for analysis.

1. The 2D image has 8 frames captured at equal intervals from a video, arranged in a **2x4 grid** from left to right, top to bottom.

2. Object labels are numbered, with **numbers matching between the 2D and BEV images to indicate the same objects**.

---

**ScanQA Prompt**

You are now required to provide answers based on the given questions.

**Important Guidelines**

1. **When answering questions, do not reference the marks directly.** These marks are only provided to assist in understanding the layout. Your answers should refer to specific objects in the scene, not the marks.

2. When describing directions or positions, use prominent objects in the image to express spatial relationships, and do not refer to labels.

3. **Keep your answers as concise as possible.** For questions regarding color, quantity, etc., aim for **1-5 words**. For questions about spatial relationships, answers can be slightly longer but should not exceed 10 words. Do not provide any additional, irrelevant information.

**Answer Format**

1. All answers must be in lowercase. Answers should not include any punctuation marks. Any numbers mentioned must be in Arabic numerals.

2. **Please format your answers as follows: '##Q1## answer1, ##Q2## answer2, ...'.**

**Examples:**

- Question: What color table is on the left side of the cabinet?
- Answer: light brown

- Question: What is on the left of the tv?
- Answer: bicycle on floor

---

**Zero-shot Prompting**

**1. Syetem Message**: <System Prompt> + <ScanQA Prompt>

**2. User Message (image type)**: <url_for_frames> + <url_for_BEV>

**3. Query Message: 'What is the black chair in front of?'**

**Refinement Procedures**

**1. Get the answer:**          '##Q1## White board.'

**2. Remove answer format:**          'White board.'

**2. Refinement and clean the answer:**

· Remove singular and plural forms.

· Remove unnecessary adjectives.

· Remove punctuation and spaces.

· Remove uppercase and lowercase distinctions.

**Final Results: 'whiteboard'**

Figure 6: **Prompts of Closed-sourse VLMs.** We show the prompts used for GPT-4o (GPT4Scene), which consist of a system prompt and a benchmark prompt. After generating responses, we further refine them.

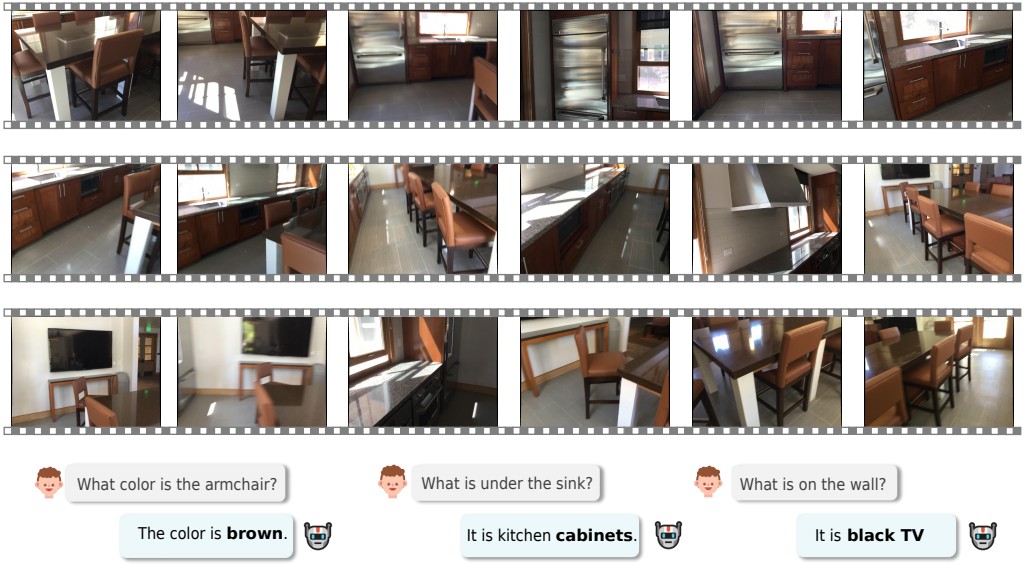

Figure 7: **Qualitive Results: Question Answering.** We provide videos without object annotations.

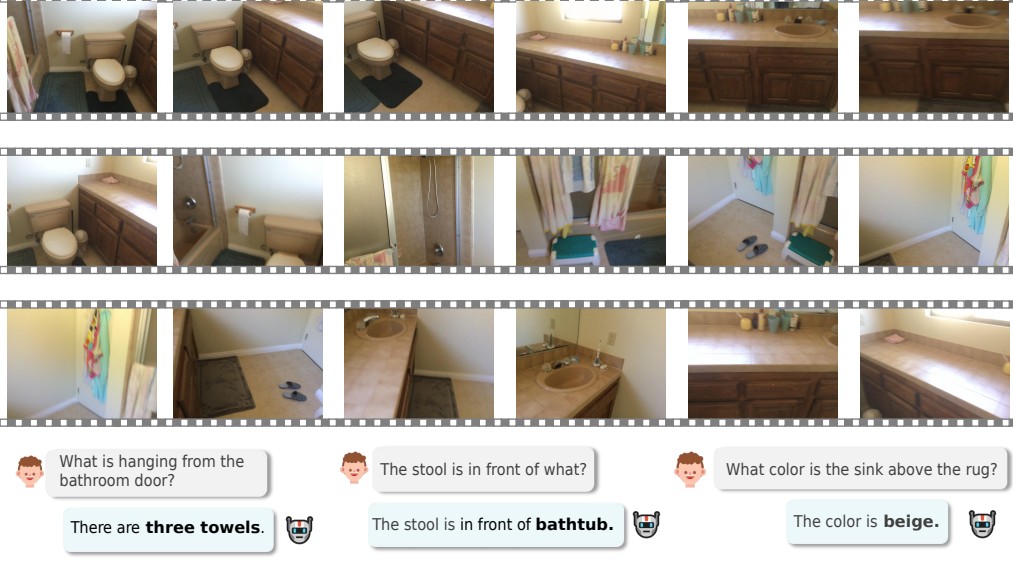

Figure 8: **Qualitive Results: Question Answering.** We provide videos without object annotations.

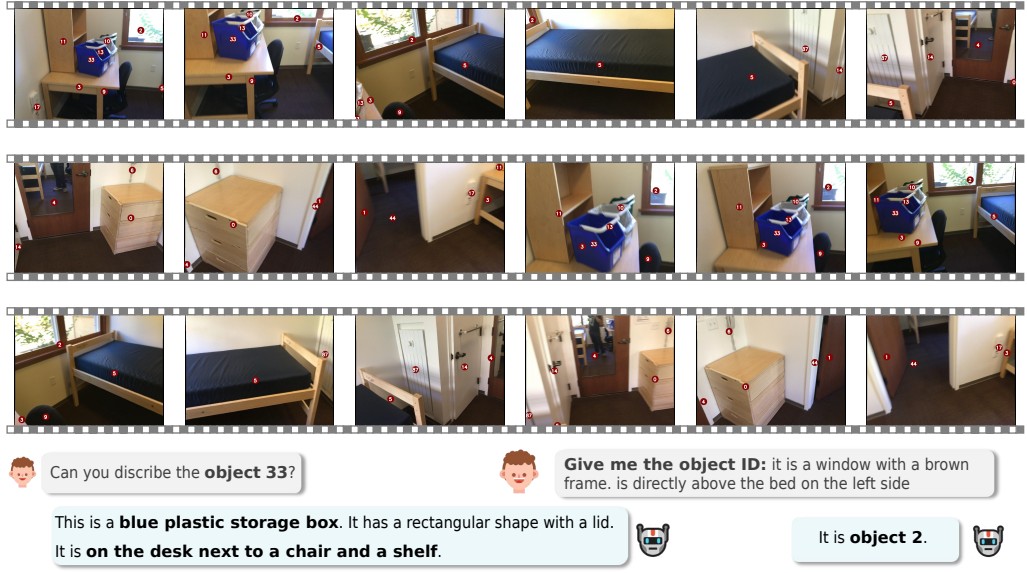

Can you discribe the **object 33**?

**Give me the object ID:** it is a window with a brown frame. is directly above the bed on the left side

This is a **blue plastic storage box**. It has a rectangular shape with a lid. It is **on the desk next to a chair and a shelf**.

It is **object 2**.

Figure 9: **Qualitive Results: 3D Dense Caption and Visual Grounding.** We provide videos with object annotations.

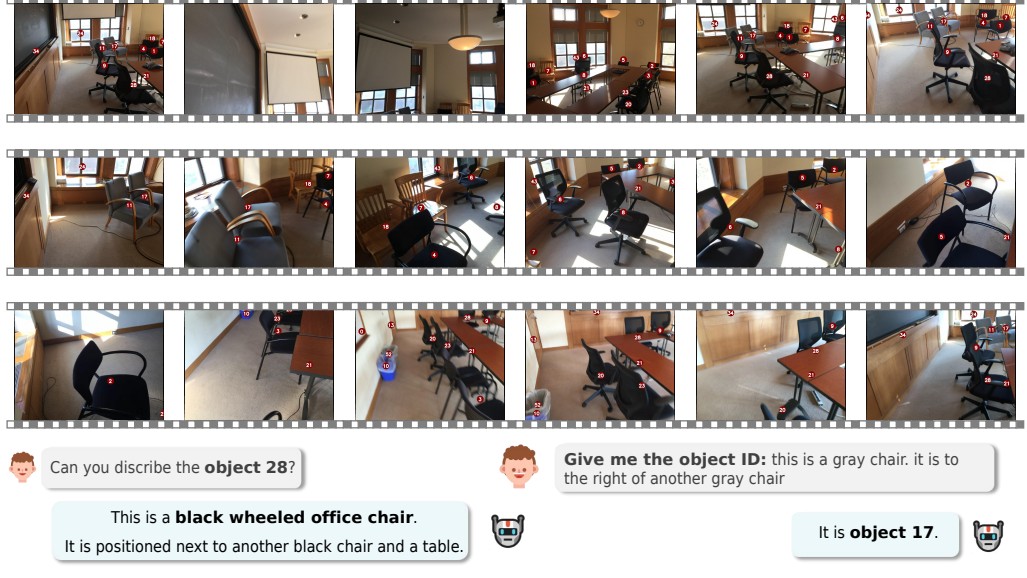

Can you discribe the **object 28**?

**Give me the object ID:** this is a gray chair. it is to the right of another gray chair

This is a **black wheeled office chair**. It is positioned next to another black chair and a table.

It is **object 17**.

Figure 10: **Qualitive Results: 3D Dense Caption and Visual Grounding.** We provide videos with object annotations.

## F    RELATED WORK

**3D indoor scene understanding.**   3D indoor scene understanding allows robots to identify object positions, structures, and relationships within indoor environments, enabling question-and-answer interactions about the scene's content. This process combines 3D perception with large language models (LLMs). 3D perception, as a foundational component, is typically trained on common indoor datasets Dai et al. (2017a); Baruch et al. (2021); Chang et al. (2017); Mao et al. (2022); Ramakrishnan et al. (2021); Wald et al. (2019); Straub et al. (2019); Deitke et al. (2022); Khanna et al. (2024); Zheng et al. (2020) using point clouds as input, supporting tasks like 3D object detection and instance segmentation Qi et al. (2019); Jiang et al. (2020); Misra et al. (2021); Schult et al. (2023); Vu et al. (2022); Wu et al. (2024); Yunhan Yang (2023). Recent advancements in 3D Vision-Language Learning (3D-VL) combine 3D perception tasks with natural language, introducing diverse textual annotations on datasets like ScanNet to support tasks such as 3D Question Answering Azuma et al. (2022); Ma et al. (2023); Ye et al. (2022); Chen et al. (2022c), 3D Dense Captioning Chen et al. (2021), and 3D Visual Grounding Chen et al. (2020); Zhang et al. (2023c); Achlioptas et al. (2020); Abdelreheem et al. (2024); Kang et al. (2025b). Initial studies focus on single 3D-VL tasks Yuan et al. (2022); Chen et al. (2023b); Wu et al. (2023); Bakr et al. (2022); He et al. (2021); Huang et al. (2022b); Jain et al. (2022); Luo et al. (2022); Yang et al. (2021); Zhao et al. (2021); Guo et al. (2023b), while recent research introduces unified models for multiple tasks Cai et al. (2022); Chen et al. (2022a). 2D vision-language pretraining (2D-VLP) has driven progress in 3D visual-language learning (3D-VL) Ha & Song (2022); Hegde et al. (2023); Peng et al. (2023); Takmaz et al. (2023); Xue et al. (2023); Zhang et al. (2022; 2023b); Xu et al. (2024a), with recent 3D-VLP Zhu et al. (2023; 2024b); Jin et al. (2023); Ding et al. (2023); Yang et al. (2024); Jia et al. (2024); Wang et al. (2024a) methods demonstrating that combining 2D visual cues with 3D point clouds enables complementary cross-modal alignment. This multimodal approach helps overcome geometric complexity and sparse annotation challenges in pure point processing.

**3D Point cloud LLMs.**   3D vision-language tasks aim to integrate 3D scene understanding with natural language processing. However, we aspire to go further by incorporating 3D content into large language models (LLMs) to achieve more natural human-computer interaction. Initially, this began with 3D point cloud LLMs. 3D point cloud LLMs take point clouds as input, enabling natural language generation and interaction in 3D scenes. Early 3D LLMs focused on object-level geometry and appearance Guo et al. (2023a); Qi et al. (2024b); Xu et al. (2024b); Qi et al. (2024a). Later, they expanded to indoor scenes, emphasizing spatial relationships among objects and overall scene features, often utilizing scene point clouds augmented with auxiliary 2D multi-view images Hong et al. (2023); Chen et al. (2024b); Wang et al. (2025c); Fu et al. (2025b); Man et al. (2024). To better capture object relationships, recent 3D LLMs decouple scene objects before feeding them into LLMs Huang et al. (2024c;a). Some approaches rely more heavily on visual inputs to determine scene context Chandrasegaran et al. (2024); Zhu et al. (2025); Liu et al. (2025). Here, we aim to explore whether pure visual inputs can better handle indoor scene understanding.

**Vision Language Models (VLMs).**   Vision Language Models (VLMs) are multimodal models that integrate visual and language processing capabilities, enabling the understanding and generation of combined image-text information. The origin of VLMs can be traced back to CLIP's 2D image-text pair pretraining Radford et al. (2021); Changpinyo et al. (2021); Schuhmann et al. (2022), which laid the foundation for incorporating LLMs. Early VLMs used attention mechanisms or Q-Former to fuse image and text modalities before inputting them to LLMs Li et al. (2022; 2023); Dai et al. (2023); Alayrac et al. (2022); Huang et al. (2023). Later, an approach emerged that directly projects image features into the LLM using an MLP Liu et al. (2023; 2024a); Li et al. (2025a); Zhu et al. (2024a); Wang et al. (2025b), achieving better performance. Building on this, VLMs expanded into visual grounding tasks Wang et al. (2024b); Peng et al. (2024); Rasheed et al. (2024); Lai et al. (2024); Dong et al. (2024); Sun et al. (2024); Chen et al. (2023a) and further to video understanding by using spatiotemporal compression to process information from long image sequences Zhang et al. (2023a); Cheng et al. (2024); Maaz et al. (2024); Luo et al. (2023); Ataallah et al. (2024); Li et al. (2024b); Song et al. (2024); Ren et al. (2024); Liu et al. (2024b). Currently, while some studies employ VLMs for indoor scene understanding, most remain either benchmark-oriented Yang et al. (2025); Li et al. (2025b) or still rely on 3D-based methods Zhang et al. (2024); Zheng et al. (2025). This fundamentally reveals that VLMs cannot directly comprehend 3D scenes, making our core mission to empower them with 3D world interpretation capabilities.

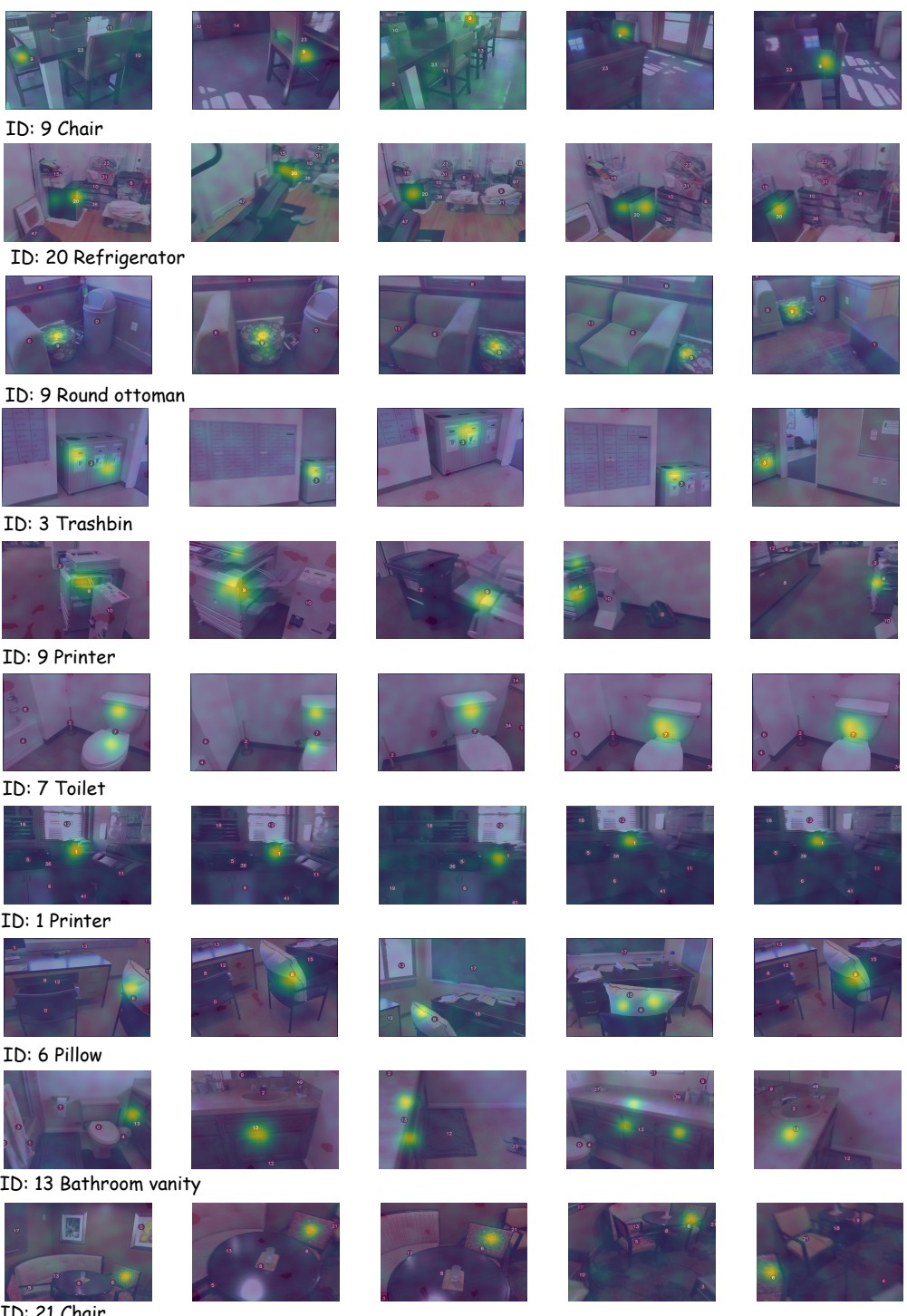

ID: 9 Chair

ID: 20 Refrigerator

ID: 9 Round ottoman

ID: 3 Trashbin

ID: 9 Printer

ID: 7 Toilet

ID: 1 Printer

ID: 6 Pillow

ID: 13 Bathroom vanity

ID: 21 Chair

Figure 11: **The heatmap after about the VLM feature.**

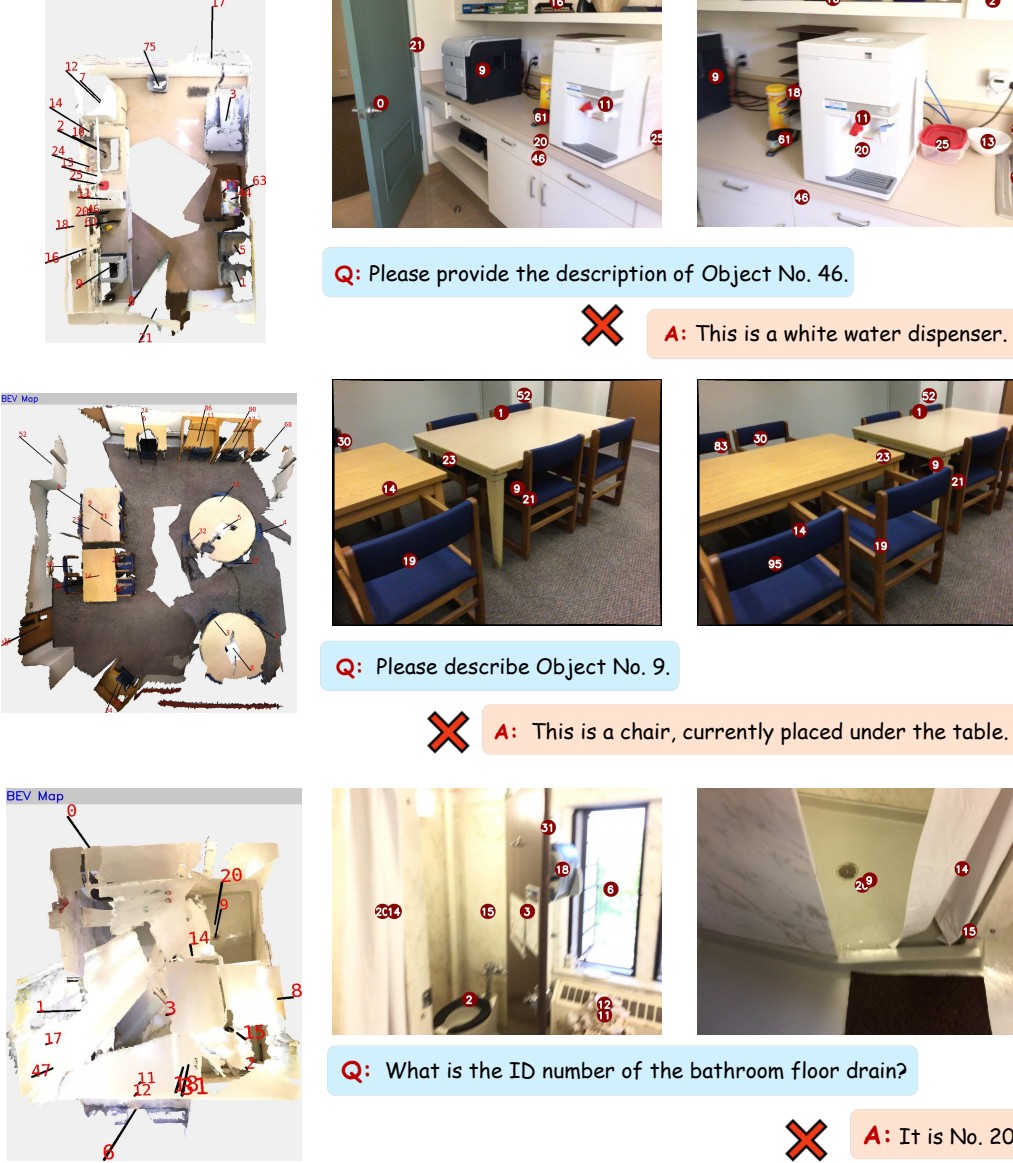

Figure 12: **Failure Cases of GPT4Scene.**

